# Metformin and Cancer Hallmarks: Molecular Mechanisms in Thyroid, Prostate and Head and Neck Cancer Models

**DOI:** 10.3390/biom12030357

**Published:** 2022-02-24

**Authors:** Mirian Galliote Morale, Rodrigo Esaki Tamura, Ileana Gabriela Sanchez Rubio

**Affiliations:** 1Department of Biological Sciences, Federal University of São Paulo, Diadema, Rua Pedro de Toledo 669, 11° Andar, São Paulo 04039-032, Brazil; morale@unifesp.br (M.G.M.); rodrigo.tamura@gmail.com (R.E.T.); 2Laboratory of Cancer Molecular Biology, Federal University of São Paulo, Rua Pedro de Toledo 669, 11° Andar, São Paulo 04039-032, Brazil; 3Thyroid Molecular Sciences Laboratory, Federal University of São Paulo, Rua Pedro de Toledo 669, 11° Andar, São Paulo 04039-032, Brazil

**Keywords:** metformin, cancer, thyroid, prostate, head and neck, hallmarks, mechanisms, in vitro

## Abstract

Metformin is the most used drug for type 2 diabetes (T2DM). Its antitumor activity has been described by clinical studies showing reduced risk of cancer development in T2DM patients, as well as management of T2DM compared with those receiving other glucose-lowering drugs. Metformin has a plethora of molecular actions in cancer cells. This review focused on in vitro data on the action mechanisms of metformin on thyroid, prostate and head and neck cancer. AMPK activation regulating specific downstream targets is a constant antineoplastic activity in different types of cancer; however, AMPK-independent mechanisms are also relevant. In vitro evidence makes it clear that depending on the type of tumor, metformin has different actions; its effects may be modulated by different cell conditions (for instance, presence of HPV infection), or it may regulate tissue-specific factors, such as the Na^+^/I^−^ symporter (NIS) and androgen receptors. The hallmarks of cancer are a set of functional features acquired by the cell during malignant development. In vitro studies show that metformin regulates almost all the hallmarks of cancer. Interestingly, metformin is one of these therapeutic agents with the potential to synergize with other chemotherapeutic agents, with low cost, low side effects and high positive consequences. Some questions are still challenging: Are metformin in vitro data able to translate from bench to bedside? Does metformin affect drug resistance? Can metformin be used as a generic anticancer drug for all types of tumors? Which are the specific actions of metformin on the peculiarities of each type of cancer? Several clinical trials are in progress or have been concluded for repurposing metformin as an anticancer drug. The continuous efforts in the field and future in vitro studies will be essential to corroborate clinical trials results and to elucidate the raised questions.

## 1. Introduction

Since the first reported use of metformin for diabetes treatment in 1957, millions of people have been taking this synthetic biguanide for type 2 diabetes management worldwide [1]. Attention has been brought to this safe and inexpensive drug for its antitumor activity from a clinical report in 2005 that associated the use of metformin with the decreased risk of cancer in patients with type 2 diabetes [2]. Since then, observational clinical studies, metanalyses and some clinical trials have associated metformin with improved cancer outcomes, reduced cancer mortality and reduced incidence across different types of cancer [3,4]. Likewise, extensive in vivo and in vitro laboratory studies have shown experimental evidence of the antineoplastic activity and underlying mechanisms of metformin in cancer, although they are not completely understood.

The action of metformin is multifactorial. In cancer, its action can be credited to inhibition of insulin mitogenic activity and activation of adenosine monophosphate-activated protein kinase (AMPK), an energy-sensing complex that regulates cellular and whole-body energy balance [5]. AMPK is activated through the tumor suppressor liver kinase B1, resulting in inhibition of the mammalian target of rapamycin pathway (mTOR), activated in many human cancers [6]. Still, metformin antitumor activity is also mediated through AMPK-independent pathways [5]. In this review, we aimed to summarize the available data on in vitro studies of the effects of metformin on human thyroid, prostate and head and neck cancers. Figure 1 summarizes the molecular pathways modulated by metformin and how it influences the cancer hallmarks [7,8,9]. 

## 2. Thyroid Cancer

Thyroid cancer (TC) is one of the most frequent endocrine cancers. TC derived from follicular cells includes differentiated thyroid cancer (DTC) with papillary (PTC) or follicular (FTC) histology (80% of cases) and undifferentiated (anaplastic-ATC) and poorly differentiated TC (1–2% of cases). Despite its good prognosis, around 10% of DTC cases progress to local and distant metastasis, lose the ability to capture radioactive iodine and no longer respond to conventional radioiodine therapy. Poorly undifferentiated TC and ATC are more aggressive, with reduced survival. Medullary thyroid cancer (MTC) derives from the parafollicular C-cells of the thyroid and is also aggressive and rare (1–2%) [10].

In human ATC and DTC cells, as well as in human thyroid primary cell cultures and rat follicular thyroid cells, metformin antimitogenic activity was correlated with induction of apoptosis and inhibition of cell growth and migration. Some variation was observed depending on the cell line [11,12,13,14,15,16,17,18,19,20,21]. The growth-inhibitory effect of metformin was also observed in thyroid cancer stem cells [11]. In Figure 1, we summarize the pathways regulated by metformin. Mechanistically, metformin increased p-AMPK, reduced mTOR phosphorylation, downregulated S6K1/S6 signaling and inhibited cyclin D1 and c-MYC through the mTOR pathway [11,12,21,22]. Besides that, metformin modulated expression of epithelial–mesenchymal transition (EMT)-related markers E-cadherin, N-cadherin and SNAIL [21]. Knockdown of the mTOR inhibitor TSC2, or treatment with rapamycin (mTOR blockade), confirmed metformin suppression of proliferation, migration and EMT of TC cells [21]. The antiproliferative activity of metformin was also observed in doxorubicin-resistant ATC cell lines, and AMPK silencing in ATC cells partially recovered phosphorylation of mTOR and cell growth inhibition by metformin [11]. 

Metformin can target thyroid cancer growth through cell metabolism. Cancer cells, despite having oxygen, often switch from mitochondrial oxidative phosphorylation (OXPHOS) to glycolysis to generate ATP, a metabolic reprogramming known as the Warburg effect [23,24]. This effect is negatively regulated by AMPK and different compounds [25,26]. In PTC, the BRAF V600E mutation altered the HIF1α-MYC-PGC-1β axis, inhibiting mitochondrial respiration and enhancing aerobic glycolysis [27]. Metformin in high glucose (20 mM) inhibited cell proliferation and in low glucose (5 mM) induced cell death, autophagy and oncosis. Cell sensitivity to metformin increased after treatment with a glycolysis inhibitor, and metformin reduced expression of the glycolytic gene PKM2, upregulated in cancer cells [22], suggesting the relationship between glucose concentration with metformin response. High expression of the metformin transporter OCT1 and mitochondrial GAPDH (mGPDH), the key enzyme connecting glycolysis with OXPHOS, was observed in TC samples compared with nontumoral samples. In FTC133 and BCPAP cell lines, metformin reduced mGPDH expression and activity and inhibited OXPHOS. Moreover, when mGPDH was silenced, metformin-mediated growth inhibition and mitochondrial respiration were reduced, while mGPDH overexpression promoted TC cell growth and sensitized cells to metformin [28]. In parallel, in PTC cells, metformin reduced 18F-fluoro-2-deoxy-d-glucose (18F-FDG) uptake and reduced levels of hexokinase-2 (HK2) and glucose transporter-1 (GLUT1), important proteins for glycolysis, showing the ability of metformin to reduce glucose metabolism [29]. These data indicate that metformin activity in TC cells depends on glucose concentration. In line with the reprogramming of cellular metabolism, metformin reduced ATP levels and mitochondrial membrane potential in ATC and PTC cells [17].

In TPC1 cells, metformin reduced p-ERK, a member of the mitogen-activated protein kinase (MAPK) family associated with proliferation and cell survival. Metformin also induced p-AKT in TPC1 and FTC236 cells with PI3K/AKT signaling activated by RET/PTEC rearrangement and PTEN mutation, respectively [12]. In a H_2_O_2_-inducible oxidative stress model, metformin attenuated H_2_O_2_ p-ERK activation, enhanced H_2_O_2_ p-AMPK expression and blocked S6K1/S6 axis, attenuating prosurvival signals and potentiating the AMPK activation in TC cells under oxidative stress [12].

In TPC-1 cells, metformin also increased expression of BIP, CHOP and caspase-12, markers of endoplasmic reticulum (ER) stress, another apoptotic mechanism [30]. Metformin ER-stress modulation was confirmed using thapsigargin, an ER-stress activator that enhanced apoptosis induced by metformin, while using the ER-stress inhibitor 4-phenylbutyrate decreased metformin-induced apoptosis [19]. The increase in metformin ER-stress was also observed in FTC 133 and BCPAP cells in low glucose medium but not in high-glucose medium [22].

Other mechanisms associated with metformin anticancer properties are the improvement in insulin resistance and reduction of serum insulin. In thyroid, the action of IGF-1/insulin is mainly by PI3K signaling promoting survival and proliferation. The PI3K pathway is the main regulator of the transcription factor FOXO1, a tumor suppressor downregulated in thyroid cancers, important for apoptosis, cell cycle, metabolism and proliferation [31]. In ATC cells, metformin reduced mRNA levels of AKT, PI3K and FOXO1 but did not modulate phosphorylation of PI3K, AKT and FOXO1, suggesting that this axis is not involved in metformin anticancer activity [18]. In turn, metformin antagonized the proliferative effect of insulin through reduction of ERK phosphorylation [11].

Metformin, independent of AMPK, also reduced expression of the multiligand transmembrane receptor LRP2 and p-JNK, a member of the MAPK family associated with survival and proliferation. The overexpression of LRP2 suppressed metformin p-JNK inhibition, suggesting that metformin inhibits the JNK pathway through LRP2 [24].

In MTC cell lines (TT and MZ-CRC-1), metformin inhibited cell growth by cell cycle arrest but did not promote apoptosis, as shown by the reduced expression of cyclin D1 without cleavage of caspase 3 and PARP after treatment [32]. In TT cell lines, metformin also reduced migration and invasion [17,21]. Metformin inhibited downstream target proteins of mTOR S6K1, S6, 4EBP1, c-MYC and cyclin D1, reduced p-ERK but not p-AKT and induced AMPK activation in TT cells [21,32]. Inhibition or silencing of AMPK did not prevent metformin downregulation of p-PS6 and partially reduced metformin inhibition of cyclin D1; thus, in TT cells, loss of AMPK activity does not completely annul the inhibitory effects of metformin on mTOR signaling, suggesting metformin acts through other pathways [32].

A concern about the first studies of metformin action on TC [12,13,32] was the use of supraphysiological doses of metformin, 10–50 mM compared to 500 to 2500 mg/d dose for diabetics. However, treatment of PTC cells with reduced concentration of metformin (0, 0.5, 1.5 and 20 mM) also reduced cell viability, increased apoptosis and activated AMPK [15]. Likewise, metformin inhibited cell proliferation and colony formation at 0.03 mM, cell migration at 0.3 M, increased apoptosis at 0.1 mM and cell cycle arrest cycle at 0.3 M on DCT, ATC and thyroid noncancer (NThyOri) cell lines [33]. These results, together with the in vivo results, suggest antitumorigenic activity of the physiological dose of metformin.

It was shown that metformin inhibits the activation of cytokine-induced nuclear factor κB (NF-κB) via AMPK activation in vascular endothelial cells [34] and that metformin/AMPK activation inhibited NF-κB signaling, upregulating IκBα in hepatocarcinoma cells [35]. In thyroid, metformin reduced the secretion of CXCL8 induced by TNF in primary cultures of normal and papillary thyroid primary cell cultures but not in TPC-1 or BCPAP cell lines [16]. CXCL8 is one of the NF-κB downstream mediators. Besides its proinflammatory properties, its expression facilitates metastasis [36]. Thus, reduced CXCL8 secretion by metformin was considered an anticancer effect, at least in classical DTC [16].

Akt-mTOR activation is essential for activation of innate immune cells and tumor-associated macrophages (TAM), important players in thyroid tumorigenesis [37,38]. It was hypothesized that metformin could modulate immune parameters associated with cancer. In a coculture model with monocytes from patients harboring PTEN mutation and thyroid cancer cells TPC1 (RET/PTC rearrangement) or FTC-133 (PTEN deficient), metformin did not alter secretion of proinflammatory cytokines from TAM induced by thyroid cancer cells. In contrast, it promoted the reduction in the anti-inflammatory cytokines IL10 and IL1-Ra. On the other hand, blockage of the mTOR pathway with rapamycin reduced the production of proinflammatory cytokines, suggesting metformin may not be effective in modulating the PTEN-mTOR axis in TAM [39].

Metformin influences metastasis and the tumor microenvironment. Whole marrow cultures treated with conditioned medium (CM) from ATC cells (ATC-CM) significantly increased production of the osteoblastic RANKL mRNA and protein, inducing osteoclast differentiation through upregulation of TRACP5b and cathepsin K markers, which were blocked by metformin treatment. Similarly, ATC-CM induced osteoclast differentiation of bone-marrow-derived monocyte/macrophage, which was also blocked by metformin treatment. The ATC-CM contained high levels of IL-6/sIL-6R and induced osteoblast RANKL production through gp130/STAT3 signaling; however, this effect was blocked by metformin in a mechanism dependent on AMPK phosphorylation, suggesting that in the tumor microenvironments, p-AMPK inhibits STAT3 phosphorylation [40].

Iodide uptake is a crucial step for radioiodine therapy for thyroid cancer treatment. In rat thyroid epithelial cells activation of AMPK reduced iodine uptake and the Na/iodide symporter (NIS) at protein and mRNA levels, while pharmacological blockage of AMPK signaling increased iodide uptake (data confirmed in an animal model) [41]. Similar results were observed in a follicular rat thyroid cell line, and AMPK modulation of NIS depended in part on the cAMP response element (CRE) present in the NIS promoter [42]. On the other hand, a recent study in ATC cells showed increased NIS mRNA and protein after metformin treatment and increased mRNA of thyroglobulin, TSHR and NKX2.1, as well as metformin acting as a demethylating agent [20]. Thus, more studies are necessary to understand the effect of metformin in iodide uptake in the thyroid cancer context.

Finally, the combination of drugs is a very interesting strategy for cancer treatment. Sorafenib is a multikinase inhibitor approved for radioiodine refractory thyroid cancer. One important issue is that it frequently promotes hard side effects, demanding a dose reduction; sorafenib combined with metformin showed a synergistic inhibition on cell growth and sphere formation in ATC cells. Additionally, metformin allowed a reduction of 25% of sorafenib dose for the same inhibitory effect [43]. In turn, the combined treatment of vemurafenib, a selective inhibitor of BRAFV600E mutant protein that constitutively activates MAPK signaling and is present in one-half of PTC and one-fourth of ATC with metformin and rapamycin, significantly reduced cell growth compared with the only one-drug treatment in 8505 (ATC) and BCPAP-vemurafenib resistant cells (PTC-BRAFV600E) [44]. In BCPAP, a similar reduction in cell growth was observed after combined treatment. Recently, synergistic effects leading to p-ERK reduction and p-AMPK increase were obtained in the combination of metformin and vemurafenib in T-238, BCPAP and HTH7 models [20].

Synergistic activity on cytotoxicity after combined treatment of metformin with gemigliptin (dipeptidyl peptidase-IV inhibitor) was observed in PTC cells through activation of AMPK and AKT. Gemigliptin increased metformin-mediated inhibition of proliferation and migration through MMP9, VCAM-1 and p-ERK reduction and p53 and p21 increase [17]. In ATC cells, combination of metformin with pioglitazone decreased expression of AKT3, DEPTOR, EIF4E, ILK, MTOR, PIK3C and PRKCA and increased expression of some tumor suppressor genes (e.g., EIF4EBP1, EIF4EBP2, PTEN) [45]. Pioglitazone is an insulin sensitizer for type 2 diabetes with therapeutic effects in mouse models of thyroid cancer due to the fusion protein (PAX8-PPARg) present in 30% of thyroid cancer [46].

## 3. Prostate Cancer

Prostate cancer is the most common cancer in men, with 1.6 million cases and around 366 thousand deaths every year. There are odds of 1:14 for men to develop prostate cancer between the ages of 0 and 79 years [47]. Androgen deprivation therapy is the major treatment strategy for advanced prostate cancer. It includes medical castration, use of antiandrogens, inhibitors of androgen biosynthesis and surgical castration [48]. Several prostate cancer cell lines have been used as study models. LNCaP, C4-2 and 22RV-1 are considered androgen-responsive cell lines, while PC3 and DU145 are androgen-independent cell lines.

Metformin has a plethora of molecular actions in prostate cancer cell lines, as shown in Figure 1. The best described is induction of AMPKα phosphorylation (Thr172), regulating downstream targets, such as acetyl-CoA carboxylase (ACC), ACC2, S6, S6K1, mTORC1 and SREBP1c [40,41,42,43,44,45,46,47,48,49,50,51,52,53]. Metformin promotes glucose fermentation (glucose uptake, lactate formation and increased glycolysis rate) in prostate cancer cell lines, inhibiting mitochondrial complex I (component of the electron transport chain) and oxygen consumption [54,55,56]. As a result, metformin decreases ATP in the cell, causing an energy defect and inhibition of lipogenesis [53]. Activation of AMPK and inhibition of mTOR has also been implicated in metformin-induced autophagy [52].

Modulation of AMPK impacts not only cell metabolism but also influences gene expression because the decrease in conversion of acetyl-CoA to malonyl-CoA causes general acetylation of proteins, including transcriptional factors (such as p65 NF-κB) and histones, altering gene expression in prostate cancer cells [57]. In fact, epigenetic regulation is a relevant factor in the metformin mechanism. It increased the expression levels of alternate histones, presenting different effects depending on the studied cell line. It increased H2AZ1 in LNCaP and C4-2, but in PC-3 and RWPE-1, it caused no significant change in H2AZ1 levels. In LNCaP and C4-2, H2A.Z occupancy in androgen receptor (AR) regions increased by metformin treatment, reducing AR expression [58]. AR is relevant in tumor development, and it was shown that AR binds to the promoter region of mIR-26a-5p, suppressing its expression and antitumor activity. There is an AMPK-AR regulatory loop in androgen-responsive cell lines. AR can also suppress AMPK signaling, and inhibition of AMPK signaling prevented metformin-induced AR decrease [59]. Metformin reduced AR and ARv7 (an AR variant correlated with worse prognosis) expression, restoring miR26a-5p expression, which suppressed enhancer of zeste homolog 2 (EZH2), a catalyst of histone-3 lysine 27 trimethylation, that suppressed gene expression and has been correlated with tumor progression [60,61]. Responsiveness to the androgen receptor is a key element in vitro and in vivo. Androgen-positive cells (DUCap and LNCaP) were the most sensitive cells to metformin, while normal epithelial cells (RWPE-1, EP-156T) were the most insensitive. The AR-positive cells showed a reduction in full-length or truncated forms of AR proteins in a metformin dose-dependent manner [62,63]. Metformin had no effect on AR degradation, stability or nuclear translocation [63]. The midline-1 (MID1) protein increased translation of the AR [64], and disruption of the MID1-α4/PP2A protein complex by metformin decreased AR protein levels and inhibited prostate cancer cell growth [62]. Metformin also induced the SMILE protein, which inhibits AR transactivation through direct interaction with AR, interfering with recruitment of the AR coactivator SRC-1 [62], showing that metformin inhibits AR through epigenetic and translational regulation or repressing its transcriptional activation.

In prostate cancer, several studies revealed that metformin mediates reduction in cell viability and arrest at the G1 phase of human prostate cancer cell lines, including DU145, PC3, LNCaP, 22Rv1, C4-2 and also canine prostate cancer cells [49,51,52,65,66,67,68,69,70,71,72,73,74,75,76,77,78,79,80], without showing evidence of an effect on normal cells (such as RWPE, PREC or P69) [52,66,77]. Due to different experimental designs, some articles did not report metformin to be an apoptotic inducer [65,66,68], and others showed different degrees of apoptotic induction [52,63,70,71,72,73,77,78,79,81].

Inhibition of proliferation by metformin may be triggered by cyclin D1 inhibition [51,65], which has been reported to be independent of AMPK activation in DU145, PC3 and LNCaP [65]. Accordingly, metformin increased sensitivity to ionizing radiation (IR) in an independent manner of AMPK activation in PC3, DU145 and 22Rv1 [51]. However, another report indicated that in PC3 cells, silencing of AMPKα prevented metformin-mediated inhibition of proliferation [50]. The combination of metformin and vitamin D3 activates AMPK phosphorylation, inhibits mOR, downregulates c-MYC and BCL-2 and causes cell cycle arrest [49]. AMPK regulation is not the sole player in metformin-induced cell cycle arrest, and different mechanisms are involved. Regulated in development and DNA damage response 1 (REDD1) was also shown to be induced by metformin in a p53-dependent manner. Silencing of REDD-1 prevented metformin-mediated cyclin D1 decrease and consequent inhibition of proliferation [67]. The Hedgehog (Hh) signaling pathway, a developmental pathway reactivated in prostate cancer cells, also regulates cyclin D1 and is inhibited by metformin [51]. Another cell cycle mediator regulated by metformin is c-MYC. Metformin promoted c-MYC phosphorylation, ubiquitination and proteasomal degradation [49,81]. The insulin-like growth factor-1 receptor (IGF-1R) activates ERK1/2 and AKT pathways and is also inhibited by metformin, reducing proliferation [82].

Induction of apoptosis by metformin is impacted by different factors, including p53 and AR status. Metformin induced expression of p53, BAX, PUMA and reduced BCL-2, hTERT and mTOR [74]. Silencing of p53 was shown to prevent apoptosis mediated by metformin in LNCaP, and ectopic expression of p53 in PC3 induced apoptosis in response to metformin [77]. Metformin regulated expression of the BCL-2 family containing BCL-2 homology domain (BH), increasing BAK and reducing BCL-2 [81]. In LNCaP, an androgen-dependent cell line, the induction of apoptosis was shown to be dependent on AMPK [66,68]. Metformin also induced apoptosis after endoplasmic reticulum (ER) stress, promoted by expression of miR-708-5p, which suppresses neuronatin (NNAT), a protein from the endoplasmic reticulum that participates in ER stress [83]. Induction of ROS production is another mechanism involved in metformin-mediated apoptosis [71,72,73].

Metformin inhibits migration and invasion of prostate cancer cells [62,70,75,76,78,82,84,85,86,87], independent of AMPK, with a more pronounced effect in AR-positive than in AR-negative cells [62]. One of the metformin targets is SUV39H1, a protein that regulates integrin signaling and was shown to increase migration when overexpressed, being downregulated by metformin [84]. Pigment epithelium-derived factor (PEDF) was shown to be induced by metformin [79]. It is an inhibitor of angiogenesis that stimulates macrophage recruitment and is usually downregulated in the tumor microenvironment [80]. Metformin was also shown to inhibit expression of IGF-1R, VEGF, FOXM1 and the phosphorylation of PI3K and Akt, which promote EMT, migration and invasion [70,82,85,86]. Metformin interferes with p-Rex-1, cAMP and CXCL12/CXCR4, inhibiting Rac1 GTPase activity and inhibiting cell motility [88]. Metformin inhibits N-cadherin, Vimentin and TWIST and upregulates E-cadherin in 22Rv1 cells in mouse xenograft models and in prostate cancer patients. Metformin also represses the levels of COX2, PGR2, TGFβ-1 and STAT3 phosphorylation, preventing STAT3-mediated expression of EMT genes, reducing migration and invasion [76,87]. Metformin was also shown to inhibit the NF-κB pathway. Metformin inhibited phosphorylation of IκBα and IKKα/β in RAS senescent cells, independent of AMPK activation [89], in a mechanism involving the axis TWIST/N-cadherin in PC3 cells. In LNCaP cells that present no detectable expression of TWIST or N-cadherin, the inhibition of p65 is dependent on AMPK [90].

LNCaP metformin-resistant cell lines showed increased proliferation, migration and invasion, expressing higher levels of EDIL2, EREG, AXL, ANAX2, CD44 and ANAX3 and lower levels of calbindin2, TPTE and IGFR1 [91]. Enriched mutations in metformin-resistant DU145 cells identified possible genes involved in metformin resistance. RAD9A and NIPSNAP1 were among the top increased genes, and RAD9A was correlated with poor prognosis in prostate cancer. High expression of RAD9A was correlated with a high proportion of Tregs, T follicular helper cells, CD8^+^ T cells, plasma cells and activated NK cells [92]. Metformin also induced expression of heme oxygenase (HO-1), a gene involved in drug resistance [72].

Combination of metformin with other drugs has been shown to increase its antitumoral activity. Salicylate, the active product of the prodrug aspirin, inhibited survival of prostate cancer cells in combination with metformin [93]. Metformin inhibited 2-deoxyglucose autophagy induction and favored apoptosis [66]. Increased apoptosis was also observed in metformin combined with bicalutamide [94], vitamin D3 [49], curcumin [74], atorvastatin [80], valproic acid [79], abiraterone and enzalutamide [61]. In fact, metformin was shown to reverse resistance to enzalutamide [78]. Metformin acted as a chemosensitizer of docetaxel, one of the main therapeutic agents against metastatic prostate cancer [95]. Combination with paclitaxel induced oxidative stress and induced apoptosis in a mitochondrial-dependent pathway [73]. Inhibitors of PLK1 combined with metformin induced the p53/REDD1 pathway and induced apoptosis [96]. Combination of metformin with simvastatin reduced proliferation, induced necrosis [97] and reduced metastatic potential [98]. Combined with quercetin synergistically repressed the VEGF/PI3K/AKT pathway [72]. Finally, metformin also increased radiosensitivity in prostate cancer cell lines [99].

Metformin reduced tumor burden in mouse xenograft models of PC3, CWR22Rv1, 22rv-1, patient-derived xenograft (PDX)-CRPC and LuCaP35CR (PDX) through oral or intraperitoneal inoculation [60,67,72,78,79,80,81,91,100,101,102,103]. One experiment showed that metformin is safe and reduced tumor growth in LNCaP, but not significantly in PC3, but increased survival in both cell lines models [104]. Metformin was shown to improve tumor oxygenation in xenograft models and improve radiotherapy responses [55]. Combination of metformin with IR reduced tumor volume and increased survival. IR alone could induce repopulation through TGF-α/EGFR activation in surviving cells, and metformin inhibited the EGFR/PI3/AKT pathway [99]. In a PDX-CRPC mouse model, metformin reduced tumor volume and reduced PLCe gene expression and reduced Notch/Hes and AR signaling [101]. In an orthotopic mouse model, metformin reduced metastasis and the CXCL12 chemotaxis agent [88]. TRAMP mice treated with metformin delayed prostate cancer progression and inhibited inflammatory infiltration by targeting the COX2/PGE2 axis, also repressing macrophage migration to the tumor microenvironment (TME) [102]. In a rat mouse model of prostate cancer, metformin was shown to have a positive therapeutic effect on renal toxicity [104], heart injury [105] and testicular damage [106]. Patients that used metformin had lower levels of PSA [107].

## 4. Head and Neck Cancer

Head and neck squamous cell carcinomas (HNSCC) are a wide range of epithelial tumors from different anatomical sites, including oral cavities such as lips, tongue and palate, for example, larynx, nasopharynx, oropharynx and hypopharynx [108]. Several factors are involved in tumor development in these sites. Correlation is well established between tobacco exposure, alcohol consumption, infection of oncogenic types of human papillomavirus (HPV) and HNSCC tumorigenesis [108]. Adding to this complexity is the treatment response that can greatly differ depending on the molecular characteristics of each tumor, and thus, drugs that have multiple effects on several molecular targets are great alternatives to improve current protocol treatments. One of these drugs is metformin.

A clinical study showed that metformin intake was related to a lower HNSCC incidence, along with an improved survival rate, decreased chance of recurrence and metastasis in HNSCC patients [109]. One mechanism that could explain this lower HNSCC incidence is the downregulation of cancer stemness and epithelial–mesenchymal transition-related genes, such as OCT4, SOX2 and NANOG, observed after metformin treatment, which could inhibit progression in cancer-initiating cells [110,111]. Unfortunately, another study showed the opposite effect depending on cell characteristics, including induction of CD44, OCT-4 and NANOG on HNSC cancer stem cells but decreased proliferation of nonstem cancer cells [112]. Therefore, studies of molecular effects of metformin in HNSCC might help understand and predict treatment outcomes in these tumors.

In HNSCC, one of the known action mechanisms of metformin is through inactivation of mTOR signaling, as shown in Figure 1. In hypopharyngeal cancer cells, p-mTOR was reduced through activation of ERK and AMPK signaling pathways by metformin, resulting in cell cycle arrest, apoptosis and autophagy triggering [113]. In oral squamous cell carcinomas (OSCC), metformin decreased expression of YES1-associated transcriptional regulator (YAP), an effector of the Hippo signaling pathway, and YAP overexpression reduced metformin inhibition on mTOR and c-MYC expression [114]. In addition, metformin inhibition of mTOR and IL6-STAT3 signaling could improve patient survival, having increased therapeutic activity in combination with CDK4/6 inhibitors [115]. The same effect was described on esophageal squamous cell carcinomas (ESCC cells), metformin-induced AMPK activation and p-mTOR, S6K1 and cyclin D1 inhibition [116].

A possible process in which metformin can act in helping treatment of apoptosis-resistant cancers is a type of nonapoptotic programmed cell death called pyroptosis. In ESCC cells, metformin was able to trigger pyroptosis by gasdermin D activation and by targeting miR-497/PELP1 axis, a pathway involved in cancer progression [117]. Additionally, on a rat model of esophageal tumor development, an equivalent result was obtained, and furthermore, metformin reduced inflammation by inhibiting iNOS, COX-2 and IL-6, which could explain the outcome of decreased incidence of precancerous lesions and ESCC in this model [116]. Using a tumor xenograft mouse model and in vitro assays, it was also demonstrated that metformin can inhibit ESCC, possibly through inhibition of NF-κB activation, IGFR1, BCL-2, caspase-9, MMP-2 and MMP-9 downregulation, in addition to E-cadherin, BAX, BIM and caspase-3 upregulation [118,119,120]. In fact, inactivation of the STAT3-BCL2 pathway in ESCC cells by metformin was previously described, establishing the relationship between metformin treatment and apoptosis and autophagy induction [121].

A specific mechanism related to smoking-driven tumors was recently described in ESCC. The cholinergic receptor nicotinic alpha 7 subunit (CHRNA7) is activated by nicotine and is a progression prognostic factor for ESCC. Metformin downregulated CHRNA7 and may be an option for treatment of nicotine-driven tumors [122].

Metformin can also modulate gene expression by epigenetic alterations. In a hypopharyngeal tumor cell, *S*-adenosylhomocysteine (SAHH) can be activated by metformin, promoting global methylation and suppressing long noncoding RNAs (lncRNAs) expression, such as SNHG7 for example. This specific lncRNA is associated with lower overall survival and treatment resistance, and metformin can sensitize hypopharyngeal tumor cells by decreasing SNHG7 expression [123].

Considering HPV-positive HNSCC cells, metformin downregulated the oncoproteins E6 and E7. A downside of this treatment was that cells were induced to enter on quiescence, evading senescence induced by other treatments [124]. Similarly, in OSCC cells, metformin inhibited proliferation but decreased cisplatin toxicity [125], an effect that can be reverted under glucose deprivation conditions [126]. In fact, glucose starvation can enhance antiproliferative effects of metformin in resistant OSCC cells and be modulated by oxygen levels [127]. In contrast, in another study, HPV-positive HNSCC cells were sensitized to ionizing radiation after combined treatment of metformin and 2-deoxy-D-glucose [128]. In nasopharyngeal carcinoma, metformin also sensitized cells to radiation through DNA repair pathway modulation, upregulation of p-ATM and p-ATR and downregulation of ATM, ATR, p95/NBS1, Rad50, DNA-PK, Ku70 and Ku80 [129]. ESCC can increase invasiveness after radiation, but metformin treatment, in addition to its antiproliferative effect, inhibits IR-induced EMT through TGF-β pathway downregulation [130].

Metformin can affect the immune microenvironment as well. In HPV-positive HNSCC, long exposure to metformin increased lymphocyte recruitment to the tumor site, with an increased CD8^+^/T-reg ratio, including the upregulation of genes related to a T-cell-inflamed state [131]. It can also interfere with M2-type tumor-associated macrophages signaling, inhibiting CCL15-CCR1-NFκB pathway activation on HNSCC and enhancing tumor susceptibility to gefitinib treatment, an EGFR inhibitor [132]. Considering tumor microenvironment, normal stromal cells have been shown to block metformin effects on OSCC cells, rescuing it from apoptosis; therefore, it is important to draw attention to the tumor as a whole and not only in individual cells, and every aspect must be considered, especially when drug combinations are being evaluated [133]. Additionally, metformin treatment can have different effects depending on HNSCC HPV status. Despite triggering CD8^+^ T cells on both, HPV-negative specimens had higher rates of apoptosis than HPV-positive ones [134].

More studies are necessary to promote understanding of metformin’s impact on cells of different site origins, with tumorigenic processes triggered by different exposures (e.g., HPV infection vs. alcohol consumption) in models that explore the tumor microenvironment interactions. 

## 5. Hallmarks of Cancer

The hallmarks of cancer define important features that modulate tumor development [7,8,9]. Every one of the hallmarks can be targeted by therapeutic agents. An efficient agent must be able to affect more than a single hallmark. In vitro evidence makes it clear that depending on the type of tumor, metformin has different actions and regulates almost all of the hallmarks. In Figure 2, we indicate genes modulated by metformin that participate in the cancer hallmarks. Metformin is one of these therapeutic agents with the potential to synergize with other chemotherapeutic agents, with low cost, low side effects and high positive consequences. Several clinical trials are in progress or have been concluded for repurposing metformin as an anticancer drug. However, additional in vitro, in vivo and clinical studies are necessary to answer open questions: Are the available data of metformin action able to translate metformin from bench to bedside? Is metformin a generic anticancer drug, or can it be used as a personalized therapy, targeting specific pathways activated by patients-specific mutation or during disease progression? Does metformin affect drug resistance? Clinical trials have strengthened the idea that metformin has therapeutic potential, and further in vitro assays will provide the detailed mechanisms that will allow these questions to be unraveled.

## Figures and Tables

**Figure 1 biomolecules-12-00357-f001:**
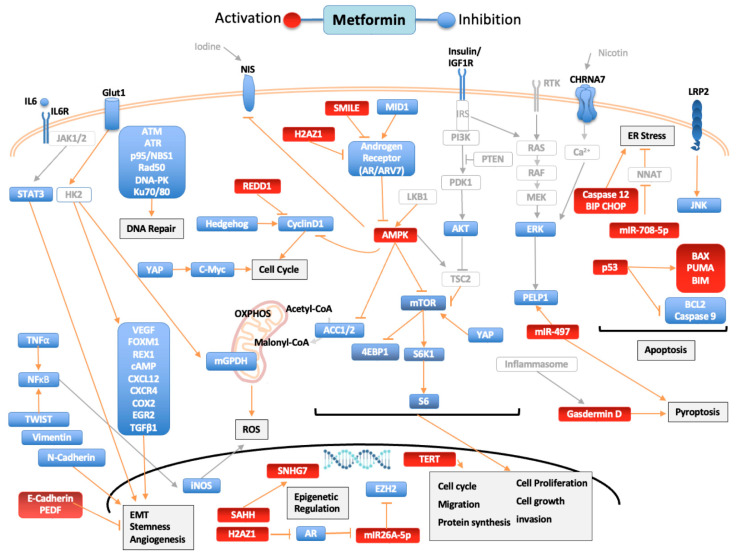
Pathways regulated by metformin in thyroid, prostate, and head and neck cancers. Blue targets show inhibited or downregulated, and red targets show activated or upregulated by metformin. Signaling molecules and arrows in gray have no direct relationship described with metformin. Acetyl-CoA carboxylase (ACC), AKT Serine/Threonine Kinase (AKT), AMPK (adenosine monophosphate-activated protein kinase), androgen receptor (AR), Androgen receptor variant 7(ARV7), Ataxia Telangiectasia Mutated (ATM), Ataxia Telangiectasia And Rad3-Related Protein (ATR), Na+/I- symporter (NIS), BCL2 Associated X, Apoptosis Regulator (BAX), Bcl-2 Interacting Mediator Of Cell Death (BIM), B-Cell CLL/Lymphoma 2 (BCL2), Binding-Immunoglobulin Protein (BIP), Cholinergic Receptor Nicotinic Alpha 7 Subunit (CHRNA7), C/EBP Homologous Protein (CHOP), cyclic AMP (cAMP), C-X-C motif chemokine receptor 4 (CXCR4), C-X-C Motif Chemokine Ligand 12 (CXCL12), Cyclooxygenase 2 (COX2), DNA-dependent protein kinase (DNA-PK), Endoplasmic reticulum (ER), Early Growth Response 2 (EGR2), Epithelial–mesenchymal transition (EMT), Eukaryotic Translation Initiation Factor 4E Binding Protein 1 (4EBP1), Extracellular Signal-Regulated Kinase 2 (ERK), Forkhead Box M1 (FOXM1), Glucose transporter 1 (Glut1), H2A.Z variant histone 1 (H2AZ1), Hexokinase-2 (HK2), Interleukin-6 (IL6), Interleukin-6 receptor (IL6R), Insulin Like Growth Factor 1 Receptor (IGF1R), Insulin Receptor Substrate (IRS), Janus Kinase (JAK), JUN N-Terminal Kinase (JNK), LDL Receptor Related Protein 2 (LRP2), X-Ray Repair Cross Complementing 6 (Ku70), X-Ray Repair Cross Complementing 5 (Ku80), Mammalian target of rapamycin (mTOR), MAPK/ERK Kinase 1 (MEK), Midline-1 (MID1), Mitochondrial GAPDH (mGPDH), MYC Proto-Oncogene, BHLH Transcription Factor (C-Myc), Neuronatin (NNAT), Nijmegen breakage syndrome 1 protein (NBS1), RAD50 Double Strand Break Repair (Rad50), Nuclear Factor Kappa B (NFκB), ZFP42 Zinc Finger (REX1), oxidative phosphorylation (OXPHOS), Pigment epithelium-derived factor (PEDF), Phosphoinositide 3-kinase (PI3K), Phosphatase And Tensin Homolog (PTEN), Proline, Glutamate And Leucine Rich Protein 1 (PELP1), Pyruvate Dehydrogenase Kinase 1 (PDK1), P53 Up-Regulated Modulator Of Apoptosis (PUMA),Raf-1 Proto-Oncogene, Serine/Threonine Kinase (RAF), Ras Proto-Oncogene GTPase (RAS), Regulated in development and DNA damage response 1 (REDD1), Ribosomal Protein S6 Kinase B1 (S6K1), Ribosomal protein S6 (S6), Small heterodimer partner–interacting leucine zipper (SMILE), Signal Transducer And Activator Of Transcription (STAT3), Transforming Growth Factor Beta 1 (TGF-β1), Tumor Necrosis Factor alpha 1 (TNF-α), S-adenosylhomocysteine (SAHH), Small Nucleolar RNA Host Gene 7 (SNHG7), Telomerase Reverse Transcriptase (TERT), TSC Complex Subunit 2 (TSC2), Twist Family BHLH Transcription Factor (TWIST), Vascular Endothelial Growth Factor (VEGF),YES1-associated transcriptional regulator (YAP).

**Figure 2 biomolecules-12-00357-f002:**
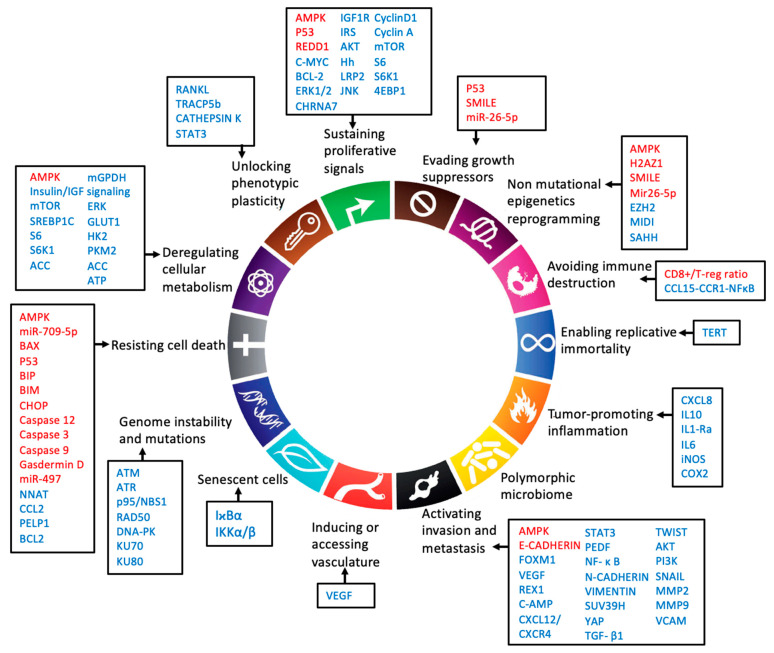
Metformin and hallmarks of cancer, genes regulated directly or indirectly by metformin in thyroid prostate and head and neck cancer. Red genes: activated; blue genes: inhibited (9 modified). Acetyl-CoA carboxylase (ACC), AKT Serine/Threonine Kinase (AKT), AMPK (adenosine monophosphate-activated protein kinase), Ataxia Telangiectasia Mutated (ATM), Adenosine triphosphate (ATP), Ataxia Telangiectasia And Rad3-Related Protein (ATR), Na+/I- symporter (NIS), BCL2 Associated X, Apoptosis Regulator (BAX), Bcl-2 Interacting Mediator Of Cell Death (BIM), B-Cell CLL/Lymphoma 2 (BCL2), Binding-Immunoglobulin Protein (BIP), Cholinergic Receptor Nicotinic Alpha 7 Subunit (CHRNA7), C/EBP Homologous Protein (CHOP), cyclic AMP (cAMP), C-C Motif Chemokine Ligand 2 (CCL2), C-C Motif Chemokine Receptor 1 (CCR1), C-X-C Motif Chemokine Ligand 8 (CXCL8), C-X-C Motif Chemokine Ligand 12 (CXCL12), C-C Motif Chemokine Ligand 15 (CC15), C-X-C motif chemokine receptor 4 (CXCR4), Cyclooxygenase 2 (COX2), DNA-dependent protein kinase (DNA-PK), Eukaryotic Translation Initiation Factor 4E Binding Protein 1 (4EBP1), Extracellular Signal-Regulated Kinase 2 (ERK), Enhancer Of Zeste 2 Polycomb Repressive Complex 2 Subunit (EZH2), Forkhead Box M1 (FOXM1), Glucose transporter 1 (Glut1), H2A.Z variant histone 1 (H2AZ1), Hexokinase-2 (HK2), Hedgehog (Hh), NFκB Inhibitor Alpha (IκBακ), Inhibitor of nuclear factor kappa-B kinase subunit alpha/beta (Iκκα/β), Interleukin-6 (IL6), Interleukin -10 (IL-10), Interleukin-1 receptor alpha (IL1-Ra), Inducible nitric oxide synthase (iNOS), Interleukin-6 receptor (IL6R), Insulin Like Growth Factor 1 Receptor (IGF1R), Insulin Receptor Substrate (IRS), JUN N-Terminal Kinase (JNK), LDL Receptor Related Protein 2 (LRP2), X-Ray Repair Cross Complementing 6 (Ku70), X-Ray Repair Cross Complementing 5 (Ku80), Mammalian target of rapamycin (mTOR), MAPK/ERK Kinase 1 (MEK), Midline-1 (MID1), Mitochondrial GAPDH (mGPDH), Matrix Metallopeptidase 2 (MMP2), Matrix Metallopeptidase (MMP9), MYC Proto-Oncogene - BHLH Transcription Factor (C-Myc), Neuronatin (NNAT), Nijmegen breakage syndrome 1 protein (NBS1), RAD50 Double Strand Break Repair (Rad50), Nuclear Factor Kappa B (NFκB), Tumor Protein P53 (P53), Pigment epithelium-derived factor (PEDF), Phosphoinositide 3-kinase (PI3K), Pyruvate Kinase M1/2 (PKM2), Proline, Glutamate And Leucine Rich Protein 1 (PELP1), Receptor Activator Of Nuclear Factor Kappa B Ligand (RANKL), Regulated in development and DNA damage response 1 (REDD1), ZFP42 Zinc Finger (REX1), Ribosomal Protein S6 Kinase B1 (S6K1), Ribosomal protein S6 (S6), Small heterodimer partner–interacting leucine zipper (SMILE), Snail Family Transcriptional Repressor 1 (SNAIL), Sterol Regulatory Element Binding Transcription Factor 1 (SREBP1C), Signal Transducer And Activator Of Transcription (STAT3), SUV39H1 Histone Lysine Methyltransferase (SUV39H), Transforming Growth Factor Beta 1 (TGF-β1), S-adenosylhomocysteine (SAHH),), Telomerase Reverse Transcriptase (TERT),), Tartrate-Resistant Acid Phosphatase 5b (TRACP5b), Twist Family BHLH Transcription Factor (TWIST), Vascular Cell Adhesion Molecule (VCAM), Vascular Endothelial Growth Factor (VEGF),YES1-associated transcriptional regulator (YAP).

## Data Availability

Not applicable.

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
