# Peer review of "Metformin and Cancer Hallmarks: Molecular Mechanisms in Thyroid, Prostate and Head and Neck Cancer Models"

_biomolecules, 2022, doi:10.3390/biom12030357_

Round 1

Reviewer 1 Report

The paper is well written providing clinically relevant valuable set of data. However, below proposed minor revisions may further increase paper visibility and overall quality of the publication.

  1. Abstract should be extended providing concluding clinically relevant statements regarding outlook answering following questions: what are the open questions and what is proposed to progress the scientific area? What are the clinically relevant conclusions and recommendations?
  2. Keeping in mind proposed additional statements (see point 1) corresponding keywords should be added to attract attention of multi-professional groups to this important publicatoin. Following items mmight be considered: personalised patient profiling, predictive approach, preventive medicine, improved individuual outcomes.
  3. Corresponding innovative concepts (see point 2) should be added presenting  "OUTLOOK" in the manuscript and within the Abstract. Below listed items and corresponding references might be useful for the authors to fulfil the task.
  • A. Metformin and mitochondial health in anti-cancer predictive and preventive approach:  Mitochondrial impairments in aetiopathology of multifactorial diseases: common origin but individual outcomes in context of 3P medicine. doi: 10.1007/s13167-021-00237-2.
  • B. Metformin and Anti-Warburg effects (mechanisms and therapy): 

    Flavonoids against the Warburg phenotype-concepts of predictive, preventive and personalised medicine to cut the Gordian knot of cancer cell metabolism. doi: 10.1007/s13167-020-00217-y.

Author Response

Reviewer #1:

The paper is well written providing clinically relevant valuable set of data. However, below proposed minor revisions may further increase paper visibility and overall quality of the publication.

Answer: We thank the reviewer comments that certainly improved the manuscript. English language spell check was performed.

1. Abstract should be extended providing concluding clinically relevant statements regarding outlook answering following questions: what are the open questions and what is proposed to progress the scientific area? What are the clinically relevant conclusions and recommendations?

Answer 1; We extended the abstract as suggested by the reviewer including open questions and information of clinical trials and relevant conclusions on in vitro studies.

2. Keeping in mind proposed additional statements (see point 1) corresponding keywords should be added to attract attention of multi-professional groups to this important publicatoin. Following items mmight be considered: personalised patient profiling, predictive approach, preventive medicine, improved individuual outcomes.

Answer 2: The focus of this review was “the in vitro studies of metformin in cancer”. This was one of the topics indicated by the Editor that were: 1) Metformin anti-cancer effects in vitro; 2) AMPK; 3) Clinical evidence of metformin anti-cancer effects; 4) Diabetes and cancer; 4) Metformin tissue availability; 5) Metformin transport in cancer cells; thus, this review focused only on the in vitro studies, we agree with the reviewer that the suggested keywords would draw more attention to other professionals in the field, however, it is out of the scope of the proposed topic and we  did not deepen on clinical aspects of metformin, even though these are really relevant topics. We mentioned in the abstract and in the conclusions the importance of the in vitro studies to corroborate and elucidate the efficacy of metformin in clinical trials for different types of cancers.

3. Corresponding innovative concepts (see point 2) should be added presenting  "OUTLOOK" in the manuscript and within the Abstract. Below listed items and corresponding references might be useful for the authors to fulfil the task.

  • A. Metformin and mitochondial health in anti-cancer predictive and preventive approach:  Mitochondrial impairments in aetiopathology of multifactorial diseases: common origin but individual outcomes in context of 3P medicine. doi: 10.1007/s13167-021-00237-2.
  • B. Metformin and Anti-Warburg effects (mechanisms and therapy): 

Flavonoids against the Warburg phenotype-concepts of predictive, preventive and personalised medicine to cut the Gordian knot of cancer cell metabolism. doi: 10.1007/s13167-020-00217-y.

Answer 3: In the abstract and in the text we included outlook considerations of metformin in cancer.  As suggested, we also included information on the Warburg effect and mitochondrial impairment, with the corresponding bibliography (doi: 10.1007/s13167-020-00217-y.and others)

Reviewer 2 Report

Thank you for the chance you gave me to read the manuscript entitled “Metformin and cancer hallmarks: molecular mechanisms in thyroid, prostate, and head and neck cancer models” (Biomolecules-1596624) by Miriam Galliote Morale et al. In this review paper, the authors present the current knowledge on the mechanisms of action of metformin on thyroid, prostate and head and neck cancers. This topic has great importance since metformin is frequently used in daily clinical practice as well as a growing number of studies have started to support its anti-cancer activity. The authors present the results of the available pre-clinical studies organizing them in 3 separate sections, one for each cancer type (thyroid, prostate and head-neck cancers). At the end, they provide the activity of metformin at these 3 cancers in the context of hallmarks of cancer. The study is well-written, has a good flow with clear messages and provides very informative figures. I think that this study in the current form satisfy the appropriate criteria for publication.

Author Response

Reviewer 2

Thank you for the chance you gave me to read the manuscript entitled “Metformin and cancer hallmarks: molecular mechanisms in thyroid, prostate, and head and neck cancer models” (Biomolecules-1596624) by Miriam Galliote Morale et al. In this review paper, the authors present the current knowledge on the mechanisms of action of metformin on thyroid, prostate and head and neck cancers. This topic has great importance since metformin is frequently used in daily clinical practice as well as a growing number of studies have started to support its anti-cancer activity. The authors present the results of the available pre-clinical studies organizing them in 3 separate sections, one for each cancer type (thyroid, prostate and head-neck cancers). At the end, they provide the activity of metformin at these 3 cancers in the context of hallmarks of cancer. The study is well-written, has a good flow with clear messages and provides very informative figures. I think that this study in the current form satisfy the appropriate criteria for publication

Answer: We appreciate a lot the comments of the reviewer, English language spell check was performed

Reviewer 3 Report

The review article by Morale et al., well summarizes the main in vitro studies on metformin and three cancer models. The authors analyze in depth the most recent literature on the topic, explaining in detail the mechanisms underlying the genes regulated directly or indirectly by metformin.

I suggest to cite the recent articles on the effect of altered cell metabolism and oxygen supply on papillary thyroid cancer cells and oral squamous cell carcinoma cell lines.

Author Response

Reviewer 3

The review article by Morale et al., well summarizes the main in vitro studies on metformin and three cancer models. The authors analyze in depth the most recent literature on the topic, explaining in detail the mechanisms underlying the genes regulated directly or indirectly by metformin.

I suggest to cite the recent articles on the effect of altered cell metabolism and oxygen supply on papillary thyroid cancer cells and oral squamous cell carcinoma cell lines.

Answer: We appreciate the comments of the reviewer, English language spell check was performed. As suggested, we included recent papers on altered cell metabolism and oxygen supply on papillary thyroid cancer cells and oral squamous cell carcinoma cell lines.